# Advances in Low-Density Flexible Polyurethane Foams by Optimized Incorporation of High Amount of Recycled Polyol

**DOI:** 10.3390/polym13111736

**Published:** 2021-05-26

**Authors:** Gabriel Kiss, Gerlinde Rusu, Geza Bandur, Iosif Hulka, Daniel Romecki, Francisc Péter

**Affiliations:** 1Faculty of Industrial Chemistry and Environmental Engineering, University Politehnica Timișoara, C. Telbisz 6, 300001 Timișoara, Romania; gabriel.kiss@momentive.com (G.K.); geza.bandur@upt.ro (G.B.); francisc.peter@upt.ro (F.P.); 2Momentive Performances Materials, Carl-Duisberg-Straße 101, 51373 Leverkusen, Germany; 3Research Institute for Renewable Energies, University Politehnica Timișoara, G. Muzicescu 138, 300501 Timișoara, Romania; iosif.hulka@upt.ro; 4Ikano Industry Sp. z o.o., Magazynowa 4, 64-610 Rogoźno, Poland; daniel.romecki@ikanoindustry.pl

**Keywords:** polyurethane foam, recycled polyol, waste recovery, foam properties, optimized foam formulation

## Abstract

An industrially manufactured recycled polyol, obtained by acidolysis process, was for the first time proved to be a possible replacement of the reference fossil-based polyol in a low-density formulation suitable for industrial production of flexible polyurethane foams. The influence of increasing recycled polyol amounts on the properties of the polyurethane foam has been studied, also performing foam emission tests to evaluate the environmental impact. Using 10 pbw recycled polyol in the standard formulation, significant differences of the physical properties were not observed, but increase of the recycled polyol amount to 30 pbw led to a dramatic decrease of the foam air flow and a very tight foam. To overcome this drawback, N,N′-*bis*[3-(dimethylamino)propyl]urea was selected as tertiary amine catalyst, enabling the preservation of foam properties even at high recycled polyol level (30 pbw). Foam emission data demonstrated that this optimized foam formulation also led to an important reduction of volatile organic compounds. The results open the way for further optimization studies in low-density flexible polyurethane foam formulations, to increase the reutilization of the polyurethane waste and reduce the amount of petroleum-based raw materials.

## 1. Introduction

Synthetic organic polymers are nowadays indispensable for everyday life, but at the same time, they represent a major source of waste. Therefore, reduction of plastic waste emerged as a major goal, and several modalities have been proposed, including its conversion into fuels or in monomers able to be recycled [1]. The global polyurethane foam market is projected to grow from roughly 15 million tons in 2020 to 20 million tons by 2025, with an annual growth rate of 7.5% from 2020 to 2025. The major reasons for the development of the polyurethane foams market include growing end-use industries such as bedding and furniture, electronics, automotive and building construction, driven also by emerging economies [2]. Special properties of polyurethane foams such as comfort, insulation, resilience or light weight are driving progress factors of the polyurethane foams market. Specifically, flexible slabstock polyurethane foam comprises a third of the total global demand of polyurethane foam. One of the main raw materials to produce flexible polyurethane foams are polyether polyols, petrochemistry-based products. With a continuous expansion of the polyurethane foam market, a linear increase of the consumption of fossil resources is expected. Thus, in accordance with the global effort to reduce the utilization of fossil-based feedstock, there is a clear need to identify alternative solutions to sustain the polyurethane growth projections.

The flexible polyurethane (PU) foams are used in various applications, such as bedding, furniture, automotive and industrials. The conversion process in complex geometries of the PU foam leads to significant foam waste, which, in extreme situations, would reach up to 20% trim foam. Moreover, old furniture and mattresses are selectively collected by specialized centers, enabling the access to polyurethane foams as a new potential feedstock. An ideal situation would be to convert the whole amount of flexible polyurethane foam waste back in the life cycle, as the recycling of plastics is one of the main challenges of the present day. From this perspective, a crucial issue is the scaling-up of promising technological solutions, as described in the literature, to commercial processes running on pilot plants and industrial facilities [3].

PU foam waste can be recycled by using physical, thermochemical and chemical methods. The physical recycling methods include regrinding, rebinding, adhesive pressing, injection molding and compress molding, while the most common chemical methods are hydrolysis, acidolysis and glycolysis [4]. Degradation of the chemical structure of the polymer allows for the depolymerization of the PU waste to oligomers terminated with hydroxyl groups, further using this product as a part of polyol component for the synthesis of new polyurethane materials [5]. Efforts in reducing fossil-based materials were studied by Serrano et al. to obtain biodegradable polyurethane foams [6]. Wenqing et al. studied the recyclability of rigid polyurethane foam [7]. Deng et al. recently presented an extensive overview of thermo-chemical recycling possibilities of PU foams. TDI and mostly polyol can be recovered by using these procedures, but pure feedstock cannot be produced, so a further upgrading of the condensate is needed, together with a thermal or alternative treatment of the non-condensable [8].

Among the chemical depolymerization methods, acidolysis may attract special interest. Gadhave et al. reviewed various chemolysis processes for depolymerization of PU foams, including acidolysis, concluding that none of them was brought into industrial practice; however, acidolysis with HCl presents the advantage of carrying out the process in mild conditions (60 °C, atmospheric pressure) [9]. However, acidolysis using dicarboxylic organic acids instead of inorganic acids looks more promising. Carboxylic acids can react at high temperatures with the polyurethane chain, yielding a recyclable polyol product. Both saturated and unsaturated dicarboxylic acids were investigated as acidolysis reagents. The research group from the University of Aveiro (Portugal) had a particularly remarkable contribution in this topic, succinic acid being identified as the most valuable reagent for the acidolysis process [10,11,12]. Rigid polyurethane foams were produced at laboratory scale using up to 30% of this recycled polyol [12]. Gama et al. also indicated a reaction scheme of depolymerization via acidolysis and identified by NMR the most probable chemical species present in the acidolysis product [6]. A pilot-scale process using dicarboxylic acids was implemented in Poland to obtain polyols from post-consumer PU matrasses, suitable for production of rigid and flexible polyurethane–polyisocyanurate foams. Rigid foams for thermo insulation were identified as the most appropriate application of these recycled polyols [13].

Although several chemical methods to recover polyurethane foam waste are described in the literature and numerous patents certifying the interest of the industry were issued in this topic [14,15,16,17,18,19,20,21,22,23], the large-scale production of recycled polyol is seldom mentioned [24]. Together with the imperative environmental issues, a viable and economically feasible recycling solution should involve high recycle yield and product quality at the same level as obtained with the virgin raw materials.

Considering the high amount of polyurethane foam waste available for recovering, as well as the global effort to reduce the fossil consumption, there is a clear demand to identify new methods or improve the efficiency of the existing methods. All of this research must be accomplished in close connection with the requirements, possibilities and prospects of the polyurethane industry.

Our previous paper reported a new approach that allows for the reutilization of the whole glycolysis product for producing flexible PU foam, but the incorporation of the recycled polyol back into low-density flexible PU foams was possible only in limited amounts, up to 5% [25]. Therefore, the main aim of the present work was to identify new ways to increase the recycled polyol amount, replacing fossil-based polyols, to produce flexible polyurethane foams in low-density formulations. Using a high percentage of recycled polyol in industrial formulations will represent an important advancement in the field, leading to a sustainable circular flow of these plastics. This study should allow a better understanding of the influence of the recovered polyol use level on foam properties. Another important novelty issue is the increase of the recycled polyol content in low-density flexible polyurethane foam formulations by selection of the appropriate tertiary amine catalyst. The selection of the catalyst is essential to obtain the desired profile in reaction, foaming, flowability and foam properties. Tertiary amines, including bis-[2-(*N*,*N*-dimethylamino)-ethyl]-ether (Amine 1) and 1,4-diazabicyclo[2.2.2]octane (Amine 2), are the usual catalysts used in industrial manufacturing of polyurethane foams. These compounds have the drawback of noticeable vapor pressure, leading to high volatile organic compound (VOC) emission and associated odor problems. Therefore, a class of so-called reactive catalysts, which are incorporated in the polymer network, has been developed, but they contain N-methyl groups which, exposed to air, can lead to the formation of formaldehyde or other degradation products [26,27]. *Bis*[3-(dimethylamino)propyl]urea (Amine 3), a typical reactive catalyst, is used to control the cream and rise time during the manufacture of flexible polyurethane foams [28]. The rational design of new tertiary amine catalysts with lower amine or formaldehyde emission will remain an essential scope of the coming years, but in our work, already-applied industrial catalysts were evaluated, also to reduce the volatile organic compounds emission. The ultimate objective was to find the optimal conditions from the perspective of the raw material, allowing the best integration back into the life cycle of the industrial scale polyurethane foam production. For this reason, we demonstrated that utilization of Amine 3 as catalyst, at high amounts of recycled polyol in the polyurethane formulation, can lead to a significant improvement of the airflow, without affecting other foam properties. This could be an important milestone towards the global efforts to reduce carbon footprint, minimize plastic waste material and contribute to the circular economy efforts for this specific material.

## 2. Materials and Methods

An important aim of this work was to investigate an industrially available recycled polyol, which could allow the overall efficiency of the PU waste reutilization. For this reason, Repolyol, which was supplied by an industrial partner, was characterized by dynamic viscosity, hydroxyl number, water content and acid number, as well as by thermogravimetric analysis and infrared spectroscopy, in comparison with the petroleum-based reference polyol.

### 2.1. Instrumental Analysis Methods

#### 2.1.1. FTIR Analysis

FTIR spectra were recorded using Bruker Vertex 70 spectrometer (Bruker Daltonik GmbH, Bremen, Germany) equipped with Platinum ATR, Bruker Diamond Tip A225/Q.1., at room temperature (4.000–400 cm^−1^), with a nominal resolution of 4 cm^−1^ with 128 scans.

#### 2.1.2. Thermogravimetric Analysis (TGA)

TGA thermograms were recorded using TG 209 F1 Libra (NETZSCH-Gerätebau GmbH, Selb, Germany) thermo gravimetrical analyzer. The measurements were carried out in nitrogen atmosphere, in the temperature range 20–600 °C, heating rate of 10 °C/min. The data were processed with the Netzsch Proteus—Thermal Analysis program version 6.1.0. (NETZSCH-Geraetebau GmbH, Selb, Germany).

#### 2.1.3. Scanning Electron Microscopy (SEM)

For the microstructural analysis the samples surfaces were characterized by scanning electron microscopy (SEM: Quanta FEG 250, FEI Europe, Eindhoven, The Netherlands), using back-scattered electron detector (BSD).

### 2.2. Physical Properties of the Polyols

The main physical characteristics of the recycled and reference polyol were determined using the ASTM international standard methods: water content (%) by ASTM D4672-18, viscosity (cSt) by ASTM D4878-15, hydroxyl number (mg KOH/g) by ASTM D4274-16 and density (g/cm^3^) by ASTM D4669-18. All measurements were made in triplicate and the data given in the Results and Discussion part represent mean values.

### 2.3. Formulation of the Flexible Polyether Polyurethane Foam

The conventional polyether foam formulation was performed according to the experimental protocol presented in Table 1, where the formulation data were expressed as ranges from minimum to maximum level. This formulation was set based on the typical protocol used in industrial manufacturing of flexible polyether polyurethanes. In order to accurately highlight the formulation changes in the experiments carried out with Repolyol, these changes are mentioned in the relevant tables in Section 3. The formulation ingredients with fixed amount were kept unchanged for all experiments. In the recycling experiments, the reference polyol was gradually replaced by different amounts of recycled polyol, keeping the total polyol amount at the same value of 100 pbw. The standard tertiary amine catalyst package, consisting of Amine 1 and Amine 2 (the chemical structures are given in Table 2) at 1:3 weight ratio was fully replaced by a reactive tertiary amine catalyst (Amine 3). The amounts of catalysts were predefined for each set of experiments, targeting a similar reactivity profile. Such predefinition of the amine catalyst amount in polyurethane foam formulations is a common procedure employed by people skilled in the art. Therefore, the initial catalyst formulation presented in this work came from the usual practice and was not optimized. The novelty of this work is based on the positive effects identified by using Amine 3 in the foam formulation containing the highest possible recycled polyol content, as is later discussed.

The equipment used for the foam preparation was a standard bench mixing station (manufactured by Pendraulik Maschinen und Apparate GmbH, Springe, Germany) with variable rotation speed, equipped with a standard impeller and rate of rise system (Format Messtechnik, Karlsruhe, Germany). The raw materials used for the polyurethane foam preparation were as follows: (i) the reference polyol (a polyoxypropylene polyoxyethylene triol, with molecular weight 3500 g/mol), marketed under the commercial name Voranol 3322, obtained from Dow Chemicals (Midland, Michigan, US); (ii) the recycled polyol, obtained at industrial scale by an acidolysis process which uses flexible polyurethane foam waste (details cannot be disclosed), supplied by IKANO Industries (Rogoźno, Poland) under the name Repolyol; (iii) toluene diisocyanate (TDI 80/20), available under commercial name Lupranate T-80, a product of BASF (Ludwigshafen, Germany); (iv) Niax Silicone L-895 (a high-performance silicon stabilizer for the production of flexible slabstock foam) and Niax Stannous Octoate (a metal-based catalyst), supplied by Momentive Performance Materials (Leverkusen, Germany).

#### Amine Catalysts Used for the Synthesis of PU Foams

Three tertiary amine catalysts were used in this work, as shown in Table 2: *bis*-[2-(*N*,*N*-dimethylamino)-ethyl]-ether (Amine 1), 1,4-diazabicyclo[2.2.2]octane (Amine 2), both purchased from Sigma Aldrich, (Steinheim, Germany) and *N*,*N*′-*bis*[3-(dimethylamino)propyl]urea (Amine 3), commercial name Niax Catalyst EF-700, supplied by Momentive Performance Materials (Leverkusen, Germany).

### 2.4. Testing Methods of the Physical Properties of the Foams

Foam density was measured on 10 × 10 × 5 cm foam samples according to DIN EN ISO 845 [29]. Compression force deflection (CFD) at 40%, expressed in kPa and SAG was measured using a 10 × 10 × 5 cm foam samples, according to the ISO 3386/1 test method [30]. Foam airflow, expressed in litters per minute (L/min), was measured on 5 × 5 × 2.5 cm foam samples, according to the ISO 7231 test method [31]. The Compression Sets were assayed according to the ASTM D 3574-05 method, with the PU sample being compressed at 75% and kept in oven at 70 °C for 22 h, and then measuring the initial vs. final thickness, expressed in %. The cell structure was characterized by visual observation [32]. Emission tests were performed in accordance to VDA-278 standard test method [33].

## 3. Results and Discussion

### 3.1. Characterization of the Recycled and Reference Polyol

#### 3.1.1. Physical Properties

Table 3 shows the physical properties of the Repolyol (obtained by an acidolysis process), compared to the reference commercial polyol (Voranol 3322). It must be pointed out that the characteristics of the reference polyol fitted well in the range provided by the producer (data not shown). The first noticeable difference was the color of the products, brown for the recycled polyol and colorless for the standard polyol (Appendix A). Hydroxyl number is an important parameter, indicating the total amount if isocyanate functional groups required during the foaming process. The hydroxyl number was similar for the two polyols. A small difference from 48 to 46.9 can be easily adjusted to ensure the correct isocyanate amount. The water content in the polyol is important, specifically to properly define the water necessary to be added in the foam formulation. Table 3 shows the same water content of the two polyols. The main difference was noticed for the viscosity. A typical viscosity of 500–600 cSt defines standard polyols, while the viscosity of the recycled polyol reached 12,500 cSt. Obviously, processing a material with such high viscosity is difficult in the industrial application. However, people skilled in the art are familiar with polyols with even higher viscosities; hence, a metering system adaptation into the production environment is required to enable the industrial use of this high viscosity material. Consequently, the recycled polyol can be considered adequate for utilization in flexible PU formulations.

#### 3.1.2. FTIR Analysis

The FTIR spectra of recycled polyol and reference polyol are presented in Figure 1. The spectra for both polyols exhibit the characteristic absorption bands at 3300–3450 cm^−1^ corresponded to stretching vibration of O–H and N–H groups. In addition, an intense peak can be observed at 1090 cm^−1^, assigned to C–O stretching. The peaks at 3000–2800 cm^−1^ correspond to the alkyl C–H stretching vibration. The spectrum of the recycled polyol shows at 1720 cm^−1^ the stretching C=O vibration of the COOH ending group from the acidolysis process, as it was also reported by Gama et al. [6]. This band is less intense in the polyurethane’s spectra (Appendix A), but it is still present due to the carbonyl group of the urethane linkage, without a noticeable shift.

The FTIR spectra of the polyurethane foams prepared by using recycled and standard polyol (Appendix A) are almost similar. The absorption bands at 3200–3400 cm^−1^ are specific for –NH stretching, while those at 2867 and 2970 cm^−1^ correspond to the stretching vibrations of CH_2_ group. The peaks that can be assigned to the urethane linkage formation [12] were identified at 1714 cm^−1^ as stretching vibrations of the C=O carbonyl functional groups of urethane, and 1537 cm^−1^ for –NH bending.

#### 3.1.3. Thermogravimetric Analysis (TGA)

Figure 2a presents the TGA thermograms for recycled polyol and reference polyol. The samples present one degradation step behavior, between 260 and 435 °C. The onset temperature for the degradation process for the recycled polyol is 373.9 °C, while for the standard polyol is 318 °C. The mass loss at 120 °C, which correspond to water content is 0.05%, corresponding basically to the water content reported in Table 3. The residual mass at 600 °C is under 3% for both polyols. The main degradation step takes place at about 50 °C higher temperature than in the case of the reference polyol, but this increased thermal stability has no adverse effect on the foam properties, as is later shown.

The TGA thermograms for the polyurethane (PU) obtained with standard polyol and recycled polyol are presented in Figure 2b. Both samples follow the same two-step degradation pattern under a nitrogen atmosphere. The first degradation step has a maximum weight-loss rate at 293 °C and corresponds to the degradation of the urethane linkages within the foam, while the second degradation step has a maximum weight-loss rate at 379 °C and involves secondary degradation processes, such as degradation of the previously formed polyols [34]. These results confirmed that, by using 30% Repolyol in the polyurethane foam formulation, the thermal stability of the foam was not affected.

### 3.2. Production of Flexible Polyurethane Foams with Increasing Amounts of Recycled Polyol

In our previous work [25], we demonstrated the possibility of recovering polyester-type foam waste by various glycolysis procedures and successfully reusing it without a purification step. Despite the important outcome of full reuse of the glycolysis product, the main shortcoming was the limitation to utmost 5% recycled polyol incorporated back in the flexible polyurethane foam. Therefore, in this work, a recycled polyol obtained at the industrial scale, by acidolysis, was studied to evaluate the influence of the recycled polyol in relationship with its use level in low-density polyurethane formulations. Further, the possibilities to incorporate higher amounts of recycled polyol back in the polyurethane foam were investigated, using a selective amine catalyst already applied in the production of flexible slabstock foam, but not yet reported in connection with recycled polyols.

#### 3.2.1. Influence of the Recycled Polyol Amount

The aim of this study was to develop laboratory-scale formulations very close to those used in the current industrial practice, allowing the best performances for flexible polyether slabstock foam production, such as a fine cell structure, stable foam and good foam porosity. At the same time, the developed formulation was sensitive enough to allow identifying a possible shift of performances.

Based on the formulation presented in Table 1, Table 4 contains the properties of the foams obtained by gradual replacement of the reference polyol used in the manufacture of commercial PU foams with recycled polyol. EXP. 1 enabled 10 parts of recycled polyol, and EXP. 2 used 20 parts, while 30 parts of recycled polyol were employed in EXP. 3. Various properties have been analyzed in a back-to-back comparison relative to the reference foam obtained with the virgin polyol. The rise time was almost similar for all foams, without significant deviation when recycled polyol was used, except for EXP. 3, when a 10-second-longer rise time was observed. Foam settling was not visible for any of these experiments, indicating good foam stabilization.

The foam properties were measured after 24 h, allowing a full curing of the polyurethane foams. The average density of the reference foam was 22.5 kg/m^3^. A standard deviation of 0.5 kg/m^3^ can be considered acceptable at laboratory stage. The addition of 10 or 20 pbw of recycled polyol did not significantly influenced the foam density. However, EXP. 3 showed a slightly higher density, 23.2 kg/m^3^, still close to the accepted standard deviation. The foam hardness revealed a significant increase with the addition of recycled polyol. At 30 pbw recycled polyol the hardness was 60% higher compared to the reference. Compression set represents the difference between initial and final thickness of foams specimen exposed to 75% compression for 22 h at 70 °C. The reference foam indicated a compressing set value of 12.34%. Generally, it is preferred to have lower percentage of compression sets values, ideally similar to the reference foam. The addition of 10 pbw Repolyol (EXP.1) enabled compression sets values similar to the reference foam but a significant increase of compression sets was noticed for EXP. 2 and EXP. 3. Another critical factor to define the foam quality is the so-called foam airflow. This parameter describes the breathability of a polyurethane foam. A foam airflow of 100 L/min with a standard deviation of 20 L/min is considered optimal. The reference foam exhibited very high foam breathability, while the addition of 10 parts of recycled polyol (EXP. 1) dropped the value to 90 L/min. The addition of recycled polyol to 20 pbw further dropped the airflow to 20 L/min, while 30 parts of recycled polyol (EXP. 3) led to very high level of closed cell content, allowing only 1 L air per minute to penetrate through the cell walls. Such a low air permeability cannot be accepted in the manufacturing practice. Therefore, further optimization of the process was required in order to accomplish the main goal of this research, the incorporation of higher amounts of recycled polyol without affecting the foam properties.

The cell-structure assessment by visual observation is also presented in Table 4, where smaller numbers are assigned for finer cell structure. For this reason, EXP. 1 indicated the same cell structure as the reference foam, while EXP. 2 and EXP. 3 indicated a slightly bigger cell structure. The modifications of the morphology of the foams, induced by increasing amounts of recycled polyol, are obvious in the optical photographs presented in Figure 3, showing the images of vertical foam slices for the set of experiments presented in Table 4. The appearance of the flexible polyurethane foam is white for the reference foam, whereas a light brown foam color was observed for EXP. 1, EXP. 2 and EXP. 3. The color change is connected to the nature of recycled polyol and widely accepted within the polyurethane industry, knowing that color change of the foam surface is expected to happen over time, hence the reference foam will also suffer yellowing effect at the surface over certain time period (such an experiment was not part of the current work). Furthermore, at 30% recycled polyol (EXP. 3), the appearance of bottom holes can be easily observed, as a typical sign of over stabilization [35].

#### 3.2.2. Optimization of the Polyurethane Foam Properties by Appropriate Selection of the Tertiary Amine Catalyst

As demonstrated in the previous section, a higher amount of recycled polyol in the foam negatively impacted the airflow in the studied low-density foam formulation. Therefore, the next objective was to identify possible ways to increase the recycled polyol amount without negatively impacting the foam airflow, by selecting the suitable catalyst. For this study, three amine catalysts and their influence were studied, keeping the reference polyol:recycled polyol ratio at 70:30. The mixture of the tertiary amines *bis*-[2-(N,N-dimethylamino)-ethyl]-ether (Amine 1) and 1,4-diazabicyclo[2.2.2]octane (Amine 2), at ratio of 1:3, is the most commonly practiced amine catalyst package in the flexible PU industry, being considered the standard amine catalysts [36]. However, using it with 30% recycled polyol a very low airflow (1 L/min) was obtained (EXP. 3). To better understand the influence of Amine 1 and Amine 2, the standard formulation reported in EXP. 3 was repeated, but using solely Amine 1 (EXP. 4) or Amine 2 (EXP. 5), in the same amount. Experiments with higher levels of Amine 1 and Amine 2 were not possible at laboratory level by this formulation, due to the high activity of these catalysts that would lead to very fast cream time (the reaction initiation time, before the PU starts expanding) and impossibility to control the reaction [36]. The properties of the foams obtained in EXP. 4 and EXP. 5 (Table 5) were also not satisfactory, specifically concerning the airflow characteristic. As a possible solution for elimination of this important drawback, a third tertiary amine, *N*,*N*′-*bis*[3-(dimethylamino)propyl]urea (Amine 3) was investigated as catalyst. The utilization of Amine 3 was possible at much higher level (0.3 parts) compared to Amine 1 and Amine 2 and did not affect the cream time and other characteristics during the PU foam formulation. The utilization of Amine 3, replacing the more conventional amine catalysts Amine 1 and Amine 2, clearly affords a higher level of recycled polyol, as demonstrated in EXP. 6, when incorporation of 30 pbw recycled polyol was possible without a significant loss of the foam properties. Moreover, Amine 3 enabled high foam airflow (87 L/min), bringing this property within the aforementioned acceptable range (Table 5). The compression sets were also significantly improved for the foam made with Amine 3, reaching a value very similar to the reference foam, as shown the Table 4. With the basic amine catalysts package, similar foam properties were obtained only when the level of recycled polyol was kept at 10 parts (EXP. 2). To the best of our best knowledge, this is the first report concerning the utilization of *N*,*N*′-*bis*[3-(dimethylamino)propyl]urea as catalyst for production of low-density flexible polyurethane foam formulations with increased level of recycled polyol.

Figure 4 shows the images of vertical foam slices for the set of experiments performed in Table 5. An obvious visual difference is the presence of bottom holes for EXP 3 and EXP 4, more pronounced for the foam made with Amine 1 as sole catalyst, sign of overstabilization at the laboratory stage. The use of Amine 2 led to almost complete elimination of the bottom holes; however, the low airflow measured for this foam makes Amine 2 also unsuitable for this formulation. Contrarily, Amine 3 led to a foam with free bottom holes, as well as high airflow, clearly demonstrating a significant improvement of the foam properties at such high recycled polyol loaded in the formulation.

Kraitape and Thongpin [37] also obtained flexible PU foams (based on methylenediphenyl diisocyanate), using a commercially available recycled polyol (Infrigreen 100). They reported the enhancement of the tensile and compressive properties of the PU foams as a result of the incorporation of the recycled polyol, making it applicable for possible automotive uses. However, smaller cell size and larger cell size distribution, as well as worse shape-recovery properties were observed even at a recycled polyol content not exceeding 10%. Contrarily, our results show that the main characteristics of the flexible PU foam were maintained at 30% recycled polyol loading.

#### 3.2.3. SEM Analysis

SEM analysis is an important tool to the more comprehensive characterization of the morphology and cellular structure of flexible PU foams. Figure 5 presents the SEM images of foams obtained with the reference polyol and with 30% recycled polyol, respectively, using Amine 3 as catalyst.

As shown in Figure 5, the cellular structure dimension is very similar for both foam samples. By using 30% recycled polyol, the morphological structure of the foam was not affected. Moreover, it can be noticed a similarity regarding the cellular dimension and the fact that most of the cells are open. The airflow defines how many open cells are in the expanded foam. Due to the similar cellular morphology the airflow values for the foam obtained with the reference and recycled polyol were similar, as shown for EXP. 1 (Table 4) and EXP. 6 (Table 5), respectively. The dimension of the cells for both foams obtained with standard and recycled polyol are between 300 and 400 μm. Furthermore, the thickness of the cell struts is very similar for both foams. These results indicate that up to 30% recycled polyol can be safely used in flexible PU foam formulations.

#### 3.2.4. Foam Emission Test Assessment for Evaluation of the Environmental Impact

The utilization of flexible PU foams can create concerns about consumer safety, particularly related to airborne pollutants like volatile organic compounds (VOC), since these substances might be released from the materials used in indoor products’ manufacturing. The emission testing procedure used in this study was the VDA-278 method [33]. For the VOC analysis, the foam sample was heated at 90 °C for 30 min. The method described in this recommendation allows the analysis of substances in the boiling/elution range up to n-pentacosane (C25), with the total value being determined as VOC. To determine the fogging (FOG) value, a second sample is retained in desorption tube after the VOC analysis and reheated to 120 °C for 60 min. The FOG value represents the total substances with low volatility with retention time starting from n-tetradecane (inclusive). It is calculated as hexadecane equivalent. For this assay, the compounds in the boiling range of n-alkanes C14 to C32 were determined and analyzed by GC–MS [35].

Table 6 shows the emission results for the reference foam and the foam made from using Amine 3, a tertiary amine designed specifically to minimize or eliminate amine emissions from the polyurethane foams [38]. The total VOC reference foam (EXP. 1) was 102 ppm [33]. The total volatile organic compound generated by the catalyst package used, EXP. 1 (including Amine 1 and Amine 2, in 1:3 ratio), was 93 ppm. The foam made from recycled polyol and including Amine 3, using 30 parts of recycled polyol (EXP. 6), had a total VOC as low as 41 ppm. The differences between the catalyst package from EXP. 1 and EXP. 6 showed a VOC reduction of almost 200%, demonstrating that the use of Amine 3 enabled a significant improvement concerning the environmental impact of the polyurethane foam.

Another important characteristic which was significantly improved was contribution of the catalyst package to the VOC, as the catalyst package for EXP. 6 contributed with 32 ppm, compared to 93 ppm in case of EXP. 1. As concerns the total FOG values, they were high also in the conditions of EXP. 6 (almost double compared to the VOC values for the same foam), because the extraction was carried out at higher temperature (120 °C) compared to the VOC analysis (90 °C). In this case, although the difference between the contribution coming out from the catalyst package was not as great, a 25% reduction was still noticed when using Amine 3.

## 4. Conclusions

The recycled polyol can be used as raw material in low-density flexible polyurethane foam formulations. However, only up to 10 parts per weight incorporation of recycled polyol can be achieved without formulation changes. The increase of recycled polyol in the same formulation affects the physical properties of the foam, specifically the airflow, which was reduced from 123 L air/min to only 1 L air/min when the amount of recycled polyol was increased to 30 pbw, indicating a very high level of closed cells. Enhancement of the foam properties at high amount of recycled polyol was accomplished by replacing the standard tertiary amine catalysts with a reactive tertiary amine catalyst, N,N′-*bis*[3-(dimethylamino)propyl]urea, identified in this study as the most efficient for the production of a flexible polyurethane foam with recycled polyols. This amine catalyst enabled three-fold higher recycled polyol level compared to the standard catalyst package, without affecting the foam properties, allowing for the production of a PU foam without defects and with optimum foam airflow. The environmental impact through foam emissions was also evaluated, demonstrating that the selected amine catalyst can also generate the reduction of the emission of the total volatile organic compounds from 102 ppm (using the reference polyol) to 41 ppm, at 30 pbw of recycled polyol and Amine 3 as catalyst.

The results of this work demonstrate that recycled polyols can be successfully incorporated in low-density flexible polyurethane foams by carefully tuning the formulation and selecting the appropriate tertiary amine catalyst. Based on these promising findings, the next step will target the production of flexible polyurethane foam with an even higher level of Repolyol, ultimately aiming to fully replace the petroleum-based raw materials in certain products. Long-term characteristics will be studied, as well, to evaluate the changes color, chemical structure and mechanical properties.

## Figures and Tables

**Figure 1 polymers-13-01736-f001:**
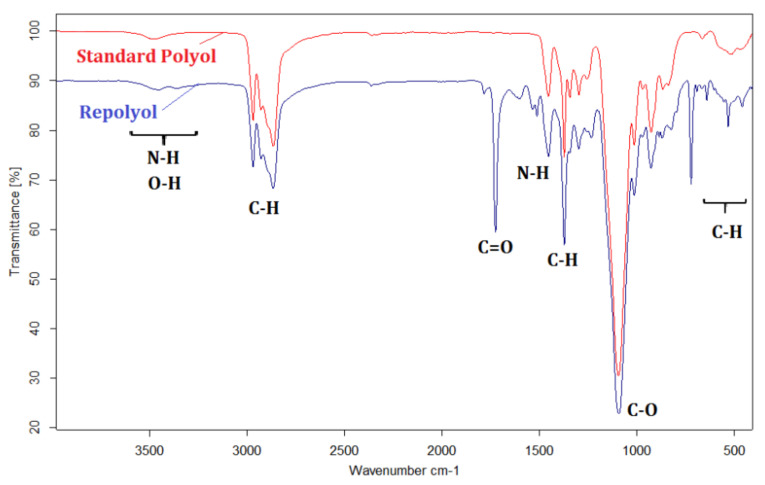
FTIR spectra of the recycled polyol (blue line, lower spectrum) and the reference polyol (red line, upper spectrum).

**Figure 2 polymers-13-01736-f002:**
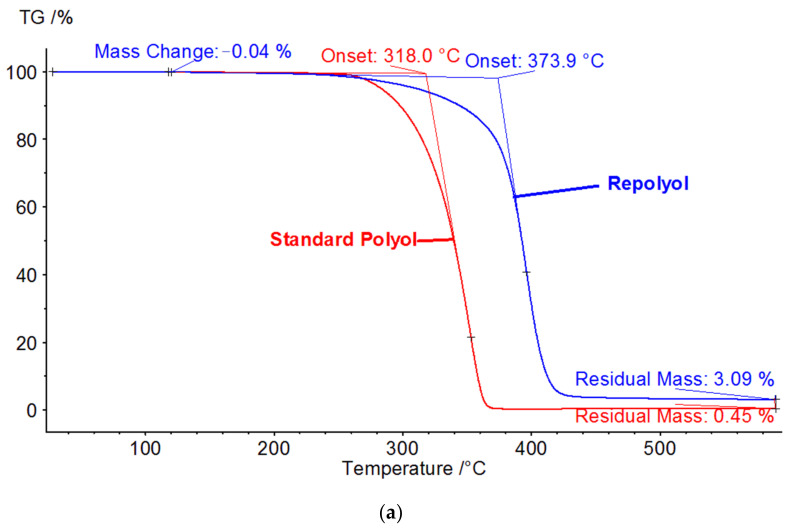
TGA thermograms of (**a**) Repolyol (blue line) and the reference polyol (red line) and (**b**) PU with Repolyol (red line) and PU with standard polyol (green line).

**Figure 3 polymers-13-01736-f003:**
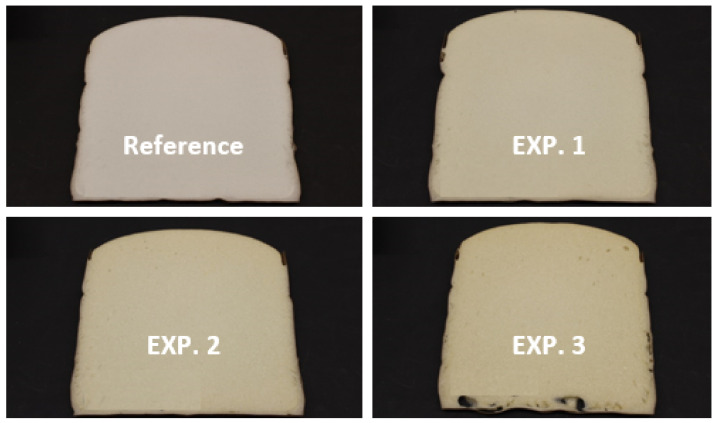
Optical photographs of the cross-section surface of the reference foam and of PU foams obtained at different reference polyol/recycled polyol ratios, 90:10 in EXP. 1, 80:20 in EXP. 2 and 70:30 in EXP 3 (the formulations are presented in Table 1 and Table 4).

**Figure 4 polymers-13-01736-f004:**
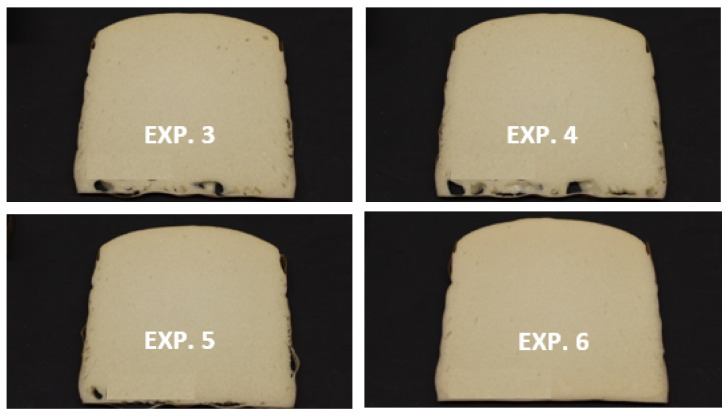
Optical photographs of the cross-section surfaces of the PU foams obtained with different tertiary amine catalysts, Amine 1 and Amine 2 at 1:3 ratio (Exp. 3), Amine 1 (Exp. 4), Amine 2 (EXP. 5) and Amine 3 (EXP. 6), at 70:30 reference polyol:Repolyol ratio, using the formulations presented in Table 5.

**Figure 5 polymers-13-01736-f005:**
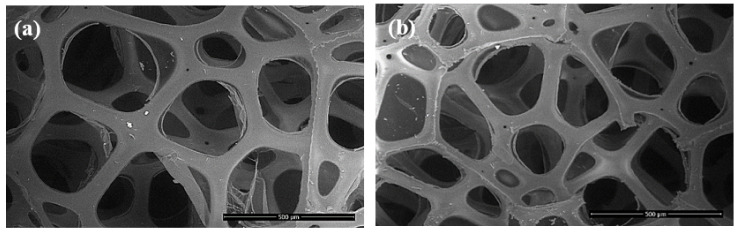
SEM images with the cellular structures of PU foams obtained with reference polyol (**a**) and 30% recycled polyol, using the formulation of EXP. 6 (**b**).

**Table 1 polymers-13-01736-t001:** Basic formulation of the flexible polyether foam.

Component	Pbw ^a^
Reference polyol (Voranol 3322)	70–100
Repolyol	0–30
Water	4.50
Niax Silicone L-895	1.00
Niax Stannous Octoate	0.16
Amine Catalyst(s) ^b^	0.06–0.30
Isocyanate index ^c^ (TDI 80/20)	108

Notes: ^a^ pbw, parts by weight. ^b^ The catalyst amount was optimized for each set of experiments, to enable similar reactivity profile. ^c^ The amount of isocyanate used in the reaction, in relation to the theoretical equivalent amount.

**Table 2 polymers-13-01736-t002:** Amine catalysts used in this study.

Catalyst	Chemical Name	Chemical Structure
Amine 1	*Bis*-[2-(N,N-dimethylamino)-ethyl]-ether	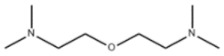
Amine 2	1,4-Diazabicyclo[2.2.2]octane	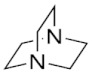
Amine 3	N,N′-*Bis*[3-(dimethylamino)propyl]urea	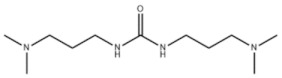

**Table 3 polymers-13-01736-t003:** Typical physical properties, assessed for the Repolyol and the reference polyol.

Characteristic	Repolyol	Reference Polyol
Color	Brown	Colorless
Water content (%)	0.05 ± 0.02	0.05 ± 0.02
Viscosity (cSt)	12,500 ± 110	550 ± 8.7
Hydroxyl number (mg KOH/g)	46.90 ± 1.02	48.00 ± 1.11
Acid number (mg KOH/g)	0.22 ± 0.01	0.05 ± 0.005

**Table 4 polymers-13-01736-t004:** The influence of the gradual substitution of the reference polyol with Repolyol on the properties of the flexible PU foams.

Formulation Changes vs. Table 1	Reference Foam	EXP. 1	EXP. 2	EXP. 3
Reference polyol (pbw)	100.0	90.0	80.0	70.0
Repolyol (pbw)	0.0	10.0	20.0	30.0
Amine 1: Amine 2 at 1:3 weight ratio (pbw)	0.06	0.06	0.06	0.06
**Foam Physical Properties**				
Rise time (s)	102	100	100	111
Foam settling (%)	-	-	-	-
Density (kg/m^3^)	22.60	22.50	22.50	23.20
Relative density	1.00	0.995	0.995	1.026
Compression set 75%, 22 h, 70 °C (%)	12.34	14.92	24.22	38.07
Hardness CFD-40% (kPa)	3.26	4.06	4.15	5.17
SAG	2.45	2.49	2.69	3.77
Airflow (L/min)	123.00	91.00	20.00	1.00
Cell structure (fine 1… coarse 8)	2	2	3	3

**Table 5 polymers-13-01736-t005:** The influence of the tertiary amine catalyst on the properties of the flexible PU foams obtained at 70:30 reference polyol:Repolyol ratio.

Formulation Changes vs. Table 1	EXP. 3	EXP. 4	EXP. 5	EXP. 6
Reference polyol (pbw)	70.0	70.0	70.0	70.0
Repolyol (pbw)	30.0	30.0	30.0	30.0
Amine 1: Amine 2, at 1:3 weight ratio (pbw)	0.06			
Amine 1 (pbw)		0.06		
Amine 2 (pbw)			0.06	
Amine 3 (pbw)				0.30
**Foam Physical Properties**				
Rise time (s)	111	111	111	99
Foam settling (%)	-	-	-	-
Density (kg/m^3^)	21.57	21.21	21.75	22.70
Relative density	1.00	0.98	1.01	1.05
Hardness CFD-40% (kPa)	5.17	5.57	4.82	4.06
Compression set 75%, 22 h, 70 °C (%)	38.07	39.55	34.21	13.40
SAG	3.77	4.19	3.36	2.67
Airflow (L/min)	1.00	1.00	1.00	87.00
Cell structure (fine 1… coarse 8)	3	3	3	2

**Table 6 polymers-13-01736-t006:** Foam emission results of foams obtained with reference polyol (EXP. 1) and using a mixture of reference polyol:Repolyol 70:30 (EXP. 6).

Formulation Changes vs. Table 1	EXP. 1	EXP. 6
Reference Polyol (pbw)	100.0	70.0
Repolyol (pbw)	-	30.0
Amine 1: Amine 2 at ratio 1:3 (pbw)	0.06	-
Amine 3 (pbw.)	-	0.30
Stannous octoate (pbw.)	0.16	0.16
**Emission Test VOC and FOG Results (ppm)**		
Total VOC	102.0	41.0
Catalyst package contribution	93.0	32.0
Total FOG	194.0	187.0
Catalyst package contribution	44.0	33.0

## Data Availability

The data presented in this study are available on request from the corresponding author.

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
