# Peer review of "Advances in Low-Density Flexible Polyurethane Foams by Optimized Incorporation of High Amount of Recycled Polyol"

_polymers, 2021, doi:10.3390/polym13111736_

Round 1

Reviewer 1 Report

The present paper reports a novel method to use recovered polyols in PU foam production. I think that it is a very interesting topic toward the global polymer recycling and circular economy. Furthermore, the paper is well presented and written and results are confirmed by reported experimentation. The Authors are encouraged to address the following minor comments.

  1. I suggest to the authors to reorganize the section 2 and 3.1 to clarify better the research process and methods. I would move section 3.1 at start of section 2, then 2.1 (1-2-3) and then 2.2 and 2.3. The section 3 could start directly with actual section 3.1.1 about characterization.
  2. Would be possible to add a picture about physical characteristic of Repolyol? (Especially regarding the different colours)
  3. Please enlarge Fig. 2 that is difficultly readable.
  4. In Table 4 could be interesting to report also the relative foam density. In fact, from fig. 3, the foam seems almost solid. How is it?
  5. The results reported in the conclusion section are too qualitative. I suggest to introduce some quantitative findings.

Author Response

The present paper reports a novel method to use recovered polyols in PU foam production. I think that it is a very interesting topic toward the global polymer recycling and circular economy. Furthermore, the paper is well presented and written and results are confirmed by reported experimentation. The Authors are encouraged to address the following minor comments.

Answer: The authors acknowledge Reviewer #1 for the comments and all help to improve this manuscript. We accepted all suggestions and the manuscript was revised accordingly. The detailed point-by-point answers are listed below.

  1. I suggest to the authors to reorganize the section 2 and 3.1 to clarify better the research process and methods. I would move section 3.1 at start of section 2, then 2.1 (1-2-3) and then 2.2 and 2.3. The section 3 could start directly with actual section 3.1.1 about characterization.

Answer: Sections 2 and 3.1 were reorganized, as requested.

  1. Would be possible to add a picture about physical characteristic of Repolyol? (Especially regarding the different colours).

Answer: In the original manuscript, the picture of Repolyol was included only in the Graphical abstract. Accepting the suggestion of Reviewer #1, this picture was also included in the Supplementary material, as Figure S1(a), together with the reference polyol, and mentioned in the manuscript (lines 219-220).

  1. Please enlarge Fig. 2 that is difficultly readable.

Answer: We apologize. Fig. 2 was modified, to make it more easily readable.

  1. In Table 4 could be interesting to report also the relative foam density. In fact, from fig. 3, the foam seems almost solid. How is it?

Answer: Indeed, the obtained foams are solid since polyurethane foams are usually low density flexible solid materials. In the revised manuscript, in Tables 4 and 5 we added, alongside the density (expressed in kg/m3), the relative density values, related to the reference foam, as suggested by Reviewer #1.

  1. The results reported in the conclusion section are too qualitative. I suggest to introduce some quantitative findings.

Answer: Thank you for this suggestion. The conclusions section was amended accordingly (lines 455-457, and 466-467).

Reviewer 2 Report

Thank you for the opportunity in reviewing the manuscript (polymers-1204935). Interesting results are reported and the manuscript was reviewed for publication in Polymers-MDPI Journal. However, the paper lacks sufficient novelty to warrant publication in Polymers-Journal. Few points why reviewer reach this decision and to improve the paper are -

  1. What do authors mean with “high amount” of recycled polyol in title of paper?
  2. The first sentence in abstract is too big. Please make it short and meaningful.
  3. Introduction is well written but insufficient. Authors need to discuss about amine 1, amine 2, and amine 3 catalyst used in the present work. How the present work is advancement from the work reported in literature especially highlighting the novelty of the present work. Only citing increasing dose of polyol from previous publication [20] by same authors is NOT sufficient to warrant publication. Why authors choose amine catalyst is NOT clear from the details?
  4. Authors must study the properties especially compressive mechanical, thermal properties of the final product. Just showing SEM, TGA or optical images is useful for nothing. Only few characterizations of the final product is main reason of rejection.
  5. In Figure 1, what authors want to demonstrate, the FTIR of standard Polyol and Repolyol is almost overlapping. The discussion part is poor. Too less interpretation and motivation of the FTIR is NOT clear. How studying FTIR related to the final product is NOT clear as well?
  6. #Figure 2 is very hard to read since the font size is too small. Similarly, the TGA of Polyol and Repolyol in Figure 2a is little different in degradation rate as a function of exposed temperature but why? It is not discussed. There the different slope of degradation in Figure 2b, what are the slope of these degradation rate is NOT calculated even though the behavior of the final products is same for both Polyol and Repolyol cases.
  7. In Figure 3 and 4, what do authors want to say from optical images? All images looks same. Even the hardness and cell structure calculated and presented in Table 4 and 5 is NOT different significantly in the experiments performed. Please cross-check?
  8. In Figure 5, only data for 30% recycled polyol using Amine 3 as catalyst is presented what about images for Amine 1 and Amine 2 as catalyst? It’s missing. Similarly, the Figure 5a and Figure 5b is almost similar. These is only slight difference in cell cavities which is NOT sufficient to warrant publication in present form of the manuscript.
  9. Lastly in table 6, only Experiment 1 and Experiment 6 data is discussed? Why others products and experiments were NOT studied? Are they NOT important? If NOT, then, why they are discussed in previous sections?

Author Response

Thank you for the opportunity in reviewing the manuscript (polymers-1204935). Interesting results are reported and the manuscript was reviewed for publication in Polymers-MDPI Journal. However, the paper lacks sufficient novelty to warrant publication in Polymers-Journal. Few points why reviewer reach this decision and to improve the paper are:

Answer: The authors acknowledge Reviewer #2 for the thorough evaluation of this manuscript and the valuable suggestions to improve it. All recommendations were considered, and we are confident that the revised form of the manuscript will be suitable for publication, also taking into account the importance of this research, which was better emphasized at the end of the Introduction part (lines 124-127). It is right that the utilization of a polyol obtained by chemical methods from polyurethane foam waste for partial replacement of virgin polyol was the subject of several publications, including our previous work, mentioned by Reviewer #2. However, the amount of the recycled polyol which can be incorporated is acritical issue, due to several difficulties associated with the manufacturing of a polyurethane foam with the same properties. In this direction, our work can be considered a breakthrough, allowing the incorporation of up to 30% recycled polyol by the optimization of a low density foam formulation which can be directly upgraded to industrial production.

  1. What do authors mean with “high amount” of recycled polyol in title of paper?

Answer: We consider that 30 parts of recycled polyol replacing the standard polyol can be considered quit a “high amount” when such a low density formulation is used (22 kg/m3). In fact, we could not find a report in the literature indicating such a high amount of recycled polyol for low density PU formulations, also preserving the foam properties. Moreover, our formulation fits well with those used in the PU industry, consequently it can be easily upscaled. Therefore, we consider that our findings represent a notable advancement in this field.

  1. The first sentence in abstract is too big. Please make it short and meaningful.

 Answer: We agree, thank you. The first sentence of the abstract was reconsidered (page 1, lines 12-14).

  1. Introduction is well written but insufficient. Authors need to discuss about amine 1, amine 2, and amine 3 catalyst used in the present work. How the present work is advancement from the work reported in literature especially highlighting the novelty of the present work. Only citing increasing dose of polyol from previous publication [20] by same authors is NOT sufficient to warrant publication. Why authors choose amine catalyst is NOT clear from the details?

Answer: Amine 1 (bis-[2-(N,N-dimethylamino)-ethyl]-ether) and Amine 2 (1,4-diazabicyclo[2.2.2]octane) are among the standard catalysts used for decades for manufacturing polyurethane foam, hence they can be considered reference tertiary amine catalysts. N,N’-bis[3-(dimethylamino)propyl]urea (Amine 3 in the text) is used to control the cream and rise time during the manufacture of polyurethane flexible foams. We considered these informations  well-known in the polymer field but following the suggestion of Reviewer #2 we added them in the revised manuscript (lines 108-119). Our studies demonstrated that utilization of Amine 3 as catalyst can lead to significant improvement of the airflow, without affecting other foam properties, at higher amounts of recycled polyol in the polyurethane formulation, compared to the standard amine catalyst (Amine 1 and Amine 2). Hence, this is a main novelty of the present work, as highlighted at the end of the Introduction part (lines 124-127). The other important improvement compared to previous publications is the higher amount of recycled polyol which can be added in the formulation, as mentioned before.

  1. Authors must study the properties especially compressive mechanical, thermal properties of the final product. Just showing SEM, TGA or optical images is useful for nothing. Only few characterizations of the final product is main reason of rejection.

Answer: In this case, we do not agree that compressive mechanical and thermal properties of the final product were scarcely presented in the manuscript, but we accept that this part must be improved. We measured the mechanical (including compressive) properties which we considered relevant for our study, the results being presented in Tables 4 and 5. Indeed, diagrams could have been more impressive (as presented by other authors in their publications) but we appreciate that the essential issue was to demonstrate that the foam obtained with recycled polyol has characteristics close to the reference foam. For low density flexible polyurethane foams, the key properties are foam density and foam hardness, expressed as compression force deflection a (CFD) at 40%. The comfort factor is another important compressive mechanical property represented by SAG, foam airflow, and cell structure. Following the observation of Reviewer #2, in the revised manuscript we added in  Tables 4 and 5 a further compression characteristic called “Compression Sets”, according to the ASTM D 3574-05 method. The PU sample is compressed at 75% and kept in oven at 70°C for 22 hours, measuring the initial vs. final thickness, expressed in %. Lower number is better, consequently we achieved excellent compressive properties also for this characteristic, for the foam obtained using the optimized formulation. The thermal properties of the foams were also studied, and the comparative results are presented in Figure 2b, showing the TGA diagrams for the polyurethane foams obtained with recycled polyol and standard polyol, respectively. In the revised manuscript, we amended the discussion part related to the thermal properties of the PU foams (lines 268-269). In addition, the SEM images provided valuable information regarding the cell structure morphology, which are also discussed in the manuscript. Based on all these reasons, we consider that in the revised manuscript the compressive mechanical and thermal properties of the final product were appropriately characterized. 

  1. In Figure 1, what authors want to demonstrate, the FTIR of standard Polyol and Repolyol is almost overlapping. The discussion part is poor. Too less interpretation and motivation of the FTIR is NOT clear. How studying FTIR related to the final product is NOT clear as well?

 Answer: In the revised manuscript, the discussion regarding the FTIR spectra was detailed both in text and in Fig. 2 (lines 237-238; 241-243; 247-252). The main objective was to demonstrate the presence of hydroxylic bonds, that will be further used to obtain polyurethane foam and the similar chemical structure of the two polyols (standard and Repolyol). The main difference between the polyol's spectra is the presence of the band at 1720 cm-1, from the carboxyl group, due to the acidolysis process. In the revised manuscript, assignment of the main peaks for the foams obtained with standard and Repolyol was also included (lines 247-252). The spectra for the polyurethane foams were added in the Supplementary materials (Fig. S2).

  1. #Figure 2 is very hard to read since the font size is too small. Similarly, the TGA of Polyol and Repolyol in Figure 2a is little different in degradation rate as a function of exposed temperature but why? It is not discussed. There the different slope of degradation in Figure 2b, what are the slope of these degradation rate is NOT calculated even though the behavior of the final products is same for both Polyol and Repolyol cases.

 Answer: Fig. 2 was modified to make it more easily readable, as also suggested by Reviewer #1. In Figure 2a the onset temperature for the degradation process of the reference polyol is lower than for the Repolyol but this thermal behaviour doesn’t affect the thermal stability of the PU foam. The thermal degradation pattern of the Polyurethane foam is two-step process as presented by other studies (reference #6, cited in the manuscript). Fig. 2 presents the inflexion temperatures for both degradation steps, which coincide with the maximum decomposition reaction rate. The thermal stability of polyurethanes depends on the crosslink density, cure conditions and the structure of the network, thus, TG analysis demonstrate that the obtained PU foam using Repolyol has similar thermal stability as the reference PU foam (lines 268-269).

  1. In Figure 3 and 4, what do authors want to say from optical images? All images looks same. Even the hardness and cell structure calculated and presented in Table 4 and 5 is NOT different significantly in the experiments performed. Please cross-check?

 Answer: Visual observation, as well as optical photographs, are among the most important assessments in flexible polyurethane foam manufacturing, allowing to understand the impact of various formulation changes. The similarity of the images is a good indication, since we targeted to not have differences concerning the cell structure size in the vertical section of a polyurethane foam. In the same time, the bottom holes are clear sign of overstabilization of the polyurethane foam, as it can be observed  in case of EXP 3, EXP 4, and very little for EXP 5 (Fig. 4). These defects are unacceptable for polyurethane foam production; hence the pictures are bringing valuable information and clearly demonstrate that a formulation containing Amine 3 (EXP 6) is appropriate,  free of bottom defects. The hardness and cell structure are expected to be similar between the foams, as our goal was to obtain characteristics similar to the reference foam using a mixture of recycled polyol and standard polyol as raw material.

  1. In Figure 5, only data for 30% recycled polyol using Amine 3 as catalyst is presented what about images for Amine 1 and Amine 2 as catalyst? It’s missing. Similarly, the Figure 5a and Figure 5b is almost similar. These is only slight difference in cell cavities which is NOT sufficient to warrant publication in present form of the manuscript.

 Answer: Figure 5 presents the SEM images  of the polyurethane foam obtained with (a) reference polyol and (b) 30% recycled polyol, using the optimized formulation. The main conclusion emerging from this figure is that the use of Amine 3  along with recycled polyol does not affect the cell structure morphology, compared to a polyurethane foam made with reference polyol. From this perspective, the similarity of the cell structures was exactly the desired result. The comparison with foams made using Amine 1 and Amine 2 as catalyst would have been irrelevant, because the airflow was in both cases about 1 l/min, making these foams unqualifiable for utilization in the desired application. The presence of bottom holes, as shown in Fig.4, also demonstrates why we considered these foams inappropriate for further investigations (including the VOC emissions).

  1. Lastly in table 6, only Experiment 1 and Experiment 6 data is discussed? Why others products and experiments were NOT studied? Are they NOT important? If NOT, then, why they are discussed in previous sections?

 Answer: In the previous sections, we carried out the optimization of the recycled polyol amount and catalyst. The results clearly demonstrated that only the formulation according to EXP. 6 can ensure foam characteristics close to that obtained with the reference polyol (EXP. 1). Therefore, it was not justified performing emission study or any further investigation if the main characteristics defining a low density flexible polyurethane foam, such as airflow, were not fulfilled.

Reviewer 3 Report

Presented work is interesting (due to application of recycling product in the synthesis of new polyurethanes foam), but should be completed in many points:

  • How foam formulation was calculated? How using of repolyol changed the formulation?
  • General scheme of acidolysis of polyurethanes should be presented in the introduction. Examplary works (e.g. 10.14314/polimery.2018.3.8,  10.1002/pol.20210066, 10.1016/j.cej.2020.125102)related to the acidolysis of polyurethanes should be described in details (mass ratio of polyurethane to acid, temperature, time and properties of obtained products)
  • Introduction should be completed by some general works: 10.1016/B978-0-12-804039-3.00014-2, 10.1016/j.wasman.2018.03.041. Introduction can also indicate the companies, which perform acidolysis process of polyurethanes
  • The analysis of polyols FTIR spectra should be extended (bands should be described in details)
  • FTIR spectra of obtained polyurethane foams should be attached and discussed in details (for example: stretching vibrations of C=O presented in urethane, urea, amide and carboxylic acid should be described in details)
  • How presence of carboxylic groups affects the preparation of polyurethane foam and properties of final polyurethanes (also in long-term using of the product)? The reaction od carboxylic groups with other components for the foam preparation should be considered
  • Hydroxyl and acid number of standard polyol and repolyol should be determined (remember to present average value with standard deviation). GPC analysis for repolyol should be performed.
  • In your opinion acidolysis of polyurethanes is a better way in comparison to glycolysis process (realized with the mass excess of glycol or with the mass excess of polyurethane) in the terms of application of obtained products during the preparation of polyurethanes foam? Please, present advantages and disadvantages
  • In my opinion, accelerated thermal aging of obtained foams should be performed. The changes of colour, chemical structure and mechanical properties should be evaluated (It is rather the proposition for further works)

Author Response

Presented work is interesting (due to application of recycling product in the synthesis of new polyurethanes foam), but should be completed in many points:

Answer: The authors acknowledge Reviewer #3 for the evaluation of the manuscript and all valuable observations and suggestions that really helped us to improve it. The point-by-point responses are given below.

  1. How foam formulation was calculated? How using of repolyol changed the formulation?

 Answer: The calculation of the formulation, presented in Table 1, followed the typical protocol used in industrial manufacturing of polyurethanes. A fixed number in Table 1 (e.g. 4.5 Pbw water) means that that it was kept constant across all experiments. Values with interval (e.g. 0-70 pbw reference polyol), mean that the amounts of the specified components were modified during the experiments, being specified in Tables 4, 5, and 6. When repolyol was used, the relative amount of reference polyol was reduced accordingly, to keep the total polyol amount at the same value of 100 Pbw. The total amount of the tertiary amine catalyst was the same in all experiments, regardless to the relative amount  of reference polyol and repolyol,  excepting EXP. 6, as it is discussed in the manuscript. The main reason to comply with an industrial protocol was to facilitate the forthcoming scaling up of our results. We added some additional explanation related to the formulation in the revised manuscript (lines 161-162, and line 167).

  1. General scheme of acidolysis of polyurethanes should be presented in the introduction. Exemplary works (e.g. 10.14314/polimery.2018.3.8, 10.1002/pol.20210066, 10.1016/j.cej.2020.125102) related to the acidolysis of polyurethanes should be described in details (mass ratio of polyurethane to acid, temperature, time and properties of obtained products)

 Answer: Thank you for this suggestion. Because the scope of our research was not to investigate the acidolysis process, repolyol being a raw material, we determined only its characteristics compared to the reference polyol. Therefore, in our opinion the inclusion of a general scheme of the acidolysis process is not justified in this work. In the same time, we agree the suggestion of Reviewer #3 that the acidolysis process must be described more detailed in the Introduction part and a paragraph with this subject was inserted in the revised manuscript, citing the references indicated by Reviewer #3 (lines 69-86, and references 11, 12, 13). Unfortunately, the requested specifications concerning the manufacturing of the repolyol used in our experiments (mass ratio of polyurethane to acid, temperature, time) are proprietary information of IKANO company, consequently they cannot be disclosed. Some data concerning such a process can be found in 10.1002/pol.20210066 (one of the publications recommended by Reviewer #3 to be included as reference) and were referred in the Introduction part (lines 81-86).

  1. Introduction should be completed by some general works: 10.1016/B978-0-12-804039-3.00014-2, 10.1016/j.wasman.2018.03.041. Introduction can also indicate the companies, which perform acidolysis process of polyurethanes.

 Answer: The Introduction was completed (line 52-55 and 59-62), citing the indicated references (references 3 and 5 in the revised manuscript). At our best knowledge IKANO is the only polyurethane foam producer performing industrial production of recycled polyol via acidolysis process. Anace GmbH company offers equipment design and manufacturing for production of PU foams also using recycled polyols (by both glycolysis and acidolysis processes) but does not mention the possible source of these polyols (https://www.anace.eu/en/polyurethane-recycling).

  1. The analysis of polyols FTIR spectra should be extended (bands should be described in details).

 Answer: The discussion relating to the FTIR spectra of the polyols was extended, as also requested by Reviewer #2 (lines 237-238; 241-243; 247-252).

  1. FTIR spectra of obtained polyurethane foams should be attached and discussed in detail (for example: stretching vibrations of C=O presented in urethane, urea, amide and carboxylic acid should be described in detail).

 Answer: In the initial manuscript, we omitted the FTIR spectra of the polyurethanes because they were almost similar (in fact this is a good result, indicating that with the PU foam obtained with recycled polyol was similar to the foam obtained with standard polyol). As suggested by Reviewer #3, in the revised manuscript we added these spectra in the Supplementary material (Fig. S2), together with the assignation of the main absorption bands, particularly the stretching vibrations of the carbonyl functional groups (lines 247-252 in the revised manuscript). 

  1. How presence of carboxylic groups affects the preparation of polyurethane foam and properties of final polyurethanes (also in long-term using of the product)? The reaction of carboxylic groups with other components for the foam preparation should be considered.

 Answer: This is a really challenging question concerning the long-term use of PU products manufactured with recycled polyol obtained by an acidolysis process. As we already mentioned, the main objective of the present study was to identify a route that will enable a higher amount of recycled polyol in the flexible foam formulation. This goal has been achieved by use of Amine 3, that enabled up to 30 parts of repolyol in the flexible polyurethane formulation, maintaining the foam properties comparable to the reference. The presence of carboxylic groups did not affect the preparation of this type of polyurethane foams as demonstrated by our results, even if some reactions with other components present in the reaction system could occur. Possibly, it may affect the hydrolytic stability of the foams and other long-term characteristics. Since long-term evaluation of the foam properties was beyond the scope of this work, for our forthcoming research we will highly consider performing the aforementioned studies.

  1. Hydroxyl and acid number of standard polyol and repolyol should be determined (remember to present average value with standard deviation). GPC analysis for repolyol should be performed.

Answer: The hydroxyl number and the acid number of standard polyol and repolyol were determined and reported in Table 3, together with other physical characteristics of interest. Standard deviation values were added as requested by Reviewer #3. In the revised manuscript, we added section 2.2 in the Methods part lines 153-157), presenting the physical characterization methods used for the polyols. GPC was not utilized, as in our opinion the characteristics presented in the Table 3 are appropriate to describe the polyols, considering that they were raw materials for our study, not produced during our experiments.

  1. In your opinion acidolysis of polyurethanes is a better way in comparison to glycolysis process (realized with the mass excess of glycol or with the mass excess of polyurethane) in the terms of application of obtained products during the preparation of polyurethanes foam? Please, present advantages and disadvantages.

 Answer: The aim of the study was to increase the amount of recycled polyol for the specific application of low density PU foams. Targeting the large-scale implementation of the process, we also needed a product which can be produced industrially, and the repolyol manufactured by acidolysis fulfilled this condition. As we did not perform a comparative study between recycled polyols obtained by different methods, we can appreciate the advantages and disadvantages of acidolysis only based on literature data. We added some consideration in this reason in the Introduction part (lines 69-74).

  1. In my opinion, accelerated thermal aging of obtained foams should be performed. The changes of colour, chemical structure and mechanical properties should be evaluated (It is rather the proposition for further works).

 Answer: The authors acknowledge Reviewer #3 for this excellent idea and suggestion. As industrial application of these results is in our view, the future studies will certainly consider performing such evaluations (as mentioned at the end of the revised manuscript (lines 472-473).

Round 2

Reviewer 2 Report

Manuscript is significantly improved. I have no further comments.

Reviewer 3 Report

In my opinion presented work can be accepted after performed corrections and complements (in the context of reviewers comments). I am waiting for further works related to the acidolysis of polyurethanes and their application in the synthesis of new materials.